# Fusion of Robotics, AI, and Thermal Imaging Technologies for Intelligent Precision Agriculture Systems

**DOI:** 10.3390/s25226844

**Published:** 2025-11-08

**Authors:** Omar Shalash, Ahmed Emad, Fares Fathy, Abdallah Alzogby, Mohamed Sallam, Eslam Naser, Mohamed El-Sayed, Esraa Khatab

**Affiliations:** 1Artificial Intelligence Research Center (AIRC), College of Engineering and Information Technology, Ajman University, Ajman P.O. Box 346, United Arab Emirates; 2College of Artificial Intelligence, Arab Academy for Science Technology and Maritime Transport, Alamein 51718, Egypt; a.ahmed0003@student.aast.edu (A.E.); f.ibrahim0329@student.aast.edu (F.F.); a.alzoghby0360@student.aast.edu (A.A.); m.h.sallam1@student.aast.edu (M.S.); e.ebra0325@student.aast.edu (E.N.); mohamedkasem@aast.edu (M.E.-S.); 3School of Mathematical and Computer Sciences, Heriot Watt University, Dubai P.O. Box 501745, United Arab Emirates; e.khatab@hw.ac.uk

**Keywords:** precision agriculture, artificial intelligence, thermal imaging, robotics, sensing

## Abstract

The world population is expected to grow to over 10 billion by 2050 and therefore impose further stress on food production. Precision agriculture has become the main approach used to enhance productivity with sustainability in agricultural production. This paper conducts a technical review of how robotics, artificial intelligence (AI), and thermal imaging (TI) technologies transform precision agriculture operations, focusing on sensing, automation, and farm decision making. Agricultural robots promote labor solutions and efficiency by utilizing their sensing devices and kinematics in planting, spraying, and harvesting. Through accurate assessment of pests/diseases and quality assurance of the harvested crops, AI and TI bring efficiency to the crop monitoring sector. Different deep learning models are employed for plant disease diagnosis and resource management, namely the VGG16 model, InceptionV3, and MobileNet; the PlantVillage, PlantDoc, and FieldPlant datasets are used respectively. To reduce crop losses, AI–TI integration enables early recognition of fluctuations caused by pests or diseases, allowing control and mitigation in good time. While the issues of cost and environmental variability (illumination, canopy moisture, and microclimate instability) are taken into consideration, the advancement in artificial intelligence, robotics technology, and combined technologies will offer sustainable solutions to the existing gaps.

## 1. Introduction

As the global population is projected to exceed 10 billion by 2050 [1], the demand for innovative and efficient methods of food production is more pressing than ever. Precision agriculture, an interdisciplinary approach that combines advanced technologies with traditional agricultural practices, has emerged as a cornerstone for addressing this challenge. By integrating machines, sensors, and data-driven management strategies, precision agriculture enhances productivity and resource efficiency by addressing variability and uncertainties inherent in agricultural systems [2].

This aligns with global initiatives such as the United Nations’ SDG 2 (Zero Hunger) and the 50 × 2030 Initiative [3], which emphasize data-driven agricultural transformation through improved farm-level information and precision technologies. Emerging technologies, in particular, with the beginning of the Industrial Revolution 4.0 [4], give a great contribution to sustainable development and improvement in quality of life by means of autonomous systems, especially in agriculture [5]. For more than three decades [6], both laboratory prototypes and field-validated agricultural robots have been developed to support operations such as planting, disease detection, spraying, and harvesting [7,8]. These robots substitute for the labor shortage, enhance efficiency, and support environmentally friendly practices to solve some of the major problems that characterize the agricultural industry: shortage of labor and inefficiency.

Recent breakthroughs in the area of AI [9,10,11,12] have transformed precision agriculture in the fields of the detection of pests and diseases in crops, their monitoring, and the management of resources. Some AI-driven technologies that have been highly transformative in the agricultural landscape include computer vision and thermal imaging (TI). TI can show subtle temperature variations that are linked to early-stage infestations and infections that are generally not visible to the human eye [13]. When employing thermal imaging, emissivity must be standardized (typically 0.95 for vegetation), and image capture should occur under consistent time-of-day, wind speed, and vapor pressure-deficit conditions to isolate disease- or stress-induced thermal signatures from general water stress effects. Therefore, AI and TI together with deep learning (DL) algorithms such as convolutional neural networks (CNNs), have provided scalable solutions to large agricultural environments in the significant improvement of their efficiencies of management of pests and diseases.

Furthermore, AI enhances the ability of data analytics automation in plant health monitoring and water stress detection in crop fields to optimize irrigation practices accordingly. DL models have been leading the race for several years in detecting rows of crops using large datasets and improving the accuracy of plant disease identification, pest attacks, and crop stress. Training of AI models for precision agriculture has been facilitated by important open-source plant disease datasets such as PlantVillage [14,15], PlantDoc [16], and FieldPlant [17]. These datasets have different characteristics that enable model training for VGG16, InceptionV3, and MobileNet in image classification tasks, particularly capturing the minutest changes in plant health conditions [18,19,20]. These models denote the capability of AI to bring a revolution to agricultural practices by reducing crop losses and therefore achieving early intervention.

The power of precision agriculture is in treating artificial intelligence (AI), robotics, and thermal imaging (TI) as complementing technologies to create autonomous intelligent farming systems. Agricultural robots that use thermal and visual sensors can capture high-resolution data on crop health and environmental conditions, while AI algorithms analyze these datasets to guide decisions such as targeted irrigation, precision spraying, or autonomous navigation [21,22,23]. Thermal imaging boosts robotic perception by detecting small canopy temperature variations that are linked to plant stress, allowing AI models to differentiate between biotic and abiotic factors with better accuracy [24,25]. This triad creates a self-optimizing ecosystem where thermal data improve AI inference, AI enhances robotic autonomy, and robotics enables scalable, real-time deployment of intelligent sensing systems.

The motivation behind this paper stems from the urgent need to address the global challenge of feeding an estimated population of over 10 billion by 2050. Traditional agricultural methods alone are insufficient to sustainably meet this rising demand for food. Labor shortages, resource inefficiencies, and climate-related uncertainties further exacerbate this challenge. By critically reviewing the latest advancements in agricultural robotics, artificial intelligence, and thermal imaging, this paper aims to highlight how integrating these technologies can revolutionize precision agriculture. The objective is to provide a review of recent studies on how smart systems can enhance crop productivity, reduce losses due to pests and diseases through early detection, and optimize resource utilization while maintaining sustainability. This comprehensive analysis will guide researchers, practitioners, and policymakers in adopting innovative, data-driven solutions to bridge existing gaps and build resilient food systems for the future. It will also discuss future work including domain adaptation that can be done in the field of precision agriculture to improve it.

## 2. Review Paper Structure

This review paper was structured systematically based on thematic areas identified through an extensive literature search. In total, approximately 62 papers related to the field of artificial intelligence (AI) and about 45 papers focusing on agricultural robotics were reviewed, covering studies published between 2015 and 2025. Major sections were defined by grouping studies using relevant keywords in online databases such as IEEE Xplore, ScienceDirect, and Google Scholar.

For the Agricultural Robots Design and System Components section, keywords such as “agricultural robot design,” “robotic end effectors,” and “robot localization in agriculture” were used.The AI for Precision Agriculture section was informed by searches using “plant disease detection deep learning,” “crop monitoring CNN models,” “PlantVillage dataset,” “PlantDoc dataset,” and related terms.For AI and Thermal Imaging, keywords included “thermal imaging plant stress,” “AI thermal pest detection,” and “post-harvest fruit quality.”

## 3. Agricultural Robots Design and System Components

Mobile robots with sensors for localization and improved path planning can navigate fields automatically. This section covers critical components for sustainable agriculture, such as gripping mechanisms, mobile robots, sensing, and navigation.

### 3.1. Robotic Grasping and Cutting Mechanisms

In recent robotic harvesting studies, various crops have been targeted with different end effectors and mechanisms for grasping and cutting. For tomatoes, Qingchun et al. [26] proposed a system comprising a vacuum cup and cutting gripper, operated by a dual-arm manipulator. This system features a double cutter and a grasping attachment and employs an open-loop control system with 3D scene reconstruction to assist in the harvesting process. For strawberries, multiple researchers developed harvesting arms utilizing a gripper with an internal container and three active fingers, where the three active fingers work alongside three passive ones. Cutting is performed using two curved blades. The proposed systems also included special features such as an inclined dropping board, a soft sponge inside the gripper, and a cover for the fingers to protect the plant during harvesting [27,28,29]. However, sweet pepper harvesting uses a mechanism with six metal fingers, which are covered in plastic. The fingers are spring-loaded, allowing them to move around the crop, while the cutting mechanism includes a plant stem fixing mechanism combined with a shaking blade. Grasp positions are recognized from a segmented 3D point cloud, with several grasping poses chosen from the point cloud data to ensure the accuracy of the harvesting process [30,31]. For lettuce, pneumatic actuators are employed for both grasping and cutting, with a blade that operates using a timing belt system. The linear action of the system is transferred to each side of the blade to ensure efficient cutting [32,33]. Lastly, eggplant harvesting relies on a simple design, with arms positioned parallel to the y-axis and the gripper closed, avoiding the need for a complex gripper mechanism to grab leaves, thus reducing the complexity of the system [26,34].

In addition, other crops have been targeted by robotic harvesting systems, each utilizing specialized end effectors and cutting mechanisms. When it comes to asparagus, a robotic arm uses two fingers and a slicing blade uses a cylindrical cam mechanism that makes it easy to perform fast arm movements. A tilt adjustment function makes the arm able to tilt up to 15 degrees, allowing it to be adaptable for use in agricultural fields [35]. Citrus harvesting is made possible by using a snake-like end effector with a biting mode, designed after the head of a snake. The system uses scissors for slicing and uses the optimized harvest postures inspired by the bionic principles to facilitate a faster harvesting process [36,37]. In ref. [38], a robotic gripper is proposed to be used for grapes; it is fused with scissors and fingers. The fingers are coated with rubber to help in defoliation, and the scissors make the slicing process very precise. The system is very diverse as it has a wide range of finger diameters (ranging from 76.2 mm to 265 mm), so it can deal with different grape sizes and shapes, helping with versatility. Lastly, pumpkins use an end effector with five fingers and seven different mechanisms, as presented in ref. [39]. Each finger has rollers and stabilizers to avoid damaging the plant, and it also includes a 60° slicing blade. This design allows for effective harvesting of pumpkins of different sizes and shapes, with a razor-edge blade that cuts through the stem with minimal force and time. The rotating end effector further helps to cut the stem efficiently. Figure 1 shows some examples of grasping and cutting end effectors, while Table 1 shows the performance of these grippers for each crop.

### 3.2. Mobile Robotic Platforms

In recent years, advancements in agricultural robotics have shifted the focus from retrofitting existing commercial tractors to developing specialized mobile platforms tailored for robotic applications. These platforms are broadly categorized into four-wheel platforms, featuring two- or four-wheel drive with steering, and tracked or six-wheel drive platforms [40]. Grimstad et al. proposed a robotic platform with the following key considerations: (1) that it can operate in wet conditions without harming the soil; (2) that it ensures affordability; and (3) it ensures flexibility through adaptable frames that allow all wheels to maintain ground contact while minimizing mechanical complexity [41].

Many other studies have investigated mobile platforms for various agricultural applications. For example, researchers in refs. [42,43,44] have explored self-propelled orchard platforms powered by four-cylinder diesel engines and equipped with hydraulic four-wheel drive for shake-and-catch apple harvesting. Further modifications have been proposed including relocating the control panel and removing worker platforms to accommodate robotic operations [42,43,44]. Articulated steer tractors have also been adapted for cotton-harvesting robots [45,46,47], while autonomous tractors have been used for heavy-duty harvesting of pumpkins and watermelons [39,48].

Among the innovative developments, a robotic mock-up model for apple harvesting was built on a Segway platform, featuring modules such as an Intel RealSense camera, a 3-DOF manipulator, and a vacuum-based end effector [49]. Another project utilized the Husky A200 platform, which, with its 68 cm width, is suitable for standard cotton row spacing. This platform is lightweight, minimizing soil compaction, and can carry loads reaching 75 kg while running on a 24 V DC battery for two to three hours under moderate cycles, while field coverage can vary depending on terrain and crop density. Additionally, it utilizes both LiDAR and GPS for row centering [50,51]. A custom-built platform by Superdroid, Inc. supported sugar snap pea harvesting with differential steering and a 30 kg capacity, moving at speeds up to 6 km/h [52].

For greenhouse applications, railed vehicle platforms have been employed for harvesting tomatoes [53,54], cherry tomatoes [26], strawberries [55,56], and sweet peppers [30]. These systems operate along guided rails, enhancing precision in confined environments. Furthermore, the TERRA-MEPP robotic platform, a tracked system, was developed for biofuel crop phenotyping, offering autonomy and multi-angle crop imaging [57]. Independently steered devices like Octinion, designed for tabletop strawberry harvesting, provide significant mobility and adaptability [58].

These platforms, summarized in Figure 2, represent a range of solutions designed to balance mobility, crop adaptability, and terrain compatibility. While progress has been made in developing semicommercial systems with advanced steering and mobility features, more research is needed to compare the suitability of different platform types for diverse agricultural tasks. Table 2 also represents an abstract overview of mobile platforms, and Table 3 shows which are the suitable terrains for each platform.

In contrast, not only unmanned ground vehicles (UGVs) are used to traverse agricultural fields, but also unmanned aerial vehicles (UAVs) such as drones. The advancement of sensor technology in the early 2000s allowed for a broader use of drones across various fields, including precision agriculture—unlike in the 1980s, when their use was exclusive to military and civil surveillance [59]. Drones can be utilized in many ways. When equipped with spraying systems, they can be used for the targeted application of pesticides, herbicides, or fertilizers. If environmental sensors are integrated, they can collect data that can later be fed into deep learning models for yield prediction, for example. Lastly, when pneumatic projection systems are mounted on drones, they can be used for seeding by propelling seeds into the soil at speeds of 200 km/h to 300 km/h, which is very useful in terrains that are difficult to traverse. Table 4 lists different types of drones along with their advantages and disadvantages in the field of precision agriculture.

**Table 2 sensors-25-06844-t002:** Overview of mobile platforms for agricultural robots [60].

Mobile Platform Type	Characteristics	Applications
Four-wheel platform with two- or four-wheel drive and two- or four-wheel steering	Lightweight, flexible frame, and suitable for wet conditions without damaging the soil structure.	Cotton harvesting, pumpkin and watermelon harvesting, apple harvesting
Tracked platform or six-wheel drives	Minimize physical effect on soil, suitable for various environmental situations.	Energy sorghum phenotyping, apple harvesting
Railed vehicle robot platform	Guided rail system for greenhouse harvesting.	Tomato harvesting, cherry tomato harvesting, strawberry harvesting, sweet pepper harvesting
Independent steering devices	Finest mobility in sloped or irregular terrain.	Strawberry picking, tomato harvesting, sugar snap pea harvesting, kiwi harvesting

**Table 3 sensors-25-06844-t003:** Task–terrain matrix for agricultural robot platforms.

Platform Type	Suitable Terrain	Crop Row Spacing	Canopy Height Limit	Typical Tasks
Railed platform	Flat and uniform surfaces (e.g., greenhouses)	Any	Low canopy	Greenhouse monitoring, fixed-path spraying
Tracked platform	Muddy or uneven ground, soft soil	≥50 cm	Medium canopy	Field navigation, harvesting in wet soil
Four-wheel platform	Firm terrain, moderate slopes	≥70 cm	High canopy	Fruit picking, transport, field inspection
Independent steering platform	Flat to moderately rough terrain	<70 cm	Low to medium canopy	Precision spraying, intra-row navigation, obstacle avoidance

**Table 4 sensors-25-06844-t004:** Comparison of different drone types in precision agriculture [59].

Drone Type	Advantages	Disadvantages
Fixed-wing drones	Large surface coverage (up to 1000 ha/day) High-altitude flight (up to 120 m legally) Efficient for large-scale mapping Long flight autonomy (1–2 h)	Limited maneuverability Requires a clear area for takeoff and landing Less suitable for detailed inspections Minimum speed required for flight
Multirotor drones	High maneuverability Hovering capability Vertical takeoff and landing Ideal for detailed inspections and targeted spraying	Limited surface coverage (50–100 ha/day) Shorter flight autonomy (20–30 min) Less efficient for mapping large areas More sensitive to strong winds
Hybrid and eVTOL drones	Combines advantages of fixed-wing and multirotor Vertical takeoff and landing Good flight autonomy (up to 1 h) Suitable for various missions	Increased mechanical complexity Higher cost May require specific training for use
Foldable-wing drones	Increased portability Easy transport and deployment Performance like fixed-wing drones Suitable for small and medium-sized farms	Potentially reduced durability due to folding mechanism May have limited payload capacity Potentially higher cost than standard models

### 3.3. Sensing and Localization

Sensing and localization are integral to the functioning of agricultural robots, allowing them to achieve critical tasks such as trajectory tracking, target localization, collision avoidance, and mapping their environment [8]. Sensors are crucial for enhancing actuation, world modeling, and decision-making, as they deliver real-time data that can be integrated and processed to support robot operations [61]. However, selecting reliable, accurate, and cost-effective sensors remains a challenge, particularly for precise localization [62,63]. To address this, many agricultural robots utilize sensor fusion techniques combining Global Navigation Satellite System (GNSS), vision-based, and Inertial Measurement Unit (IMU) data to maintain accurate positioning. While GNSS provides accurate global localization in open fields, its accuracy degrades under canopy cover or dense vegetation. Vision-based methods and IMU integration compensate for this degradation, ensuring robust localization [38,39,48,56,57,64]. Despite the variety of available sensors, their effectiveness can change across agricultural tasks. Stereo vision systems provide dense depth maps perfect for row-following and plant localization, but their performance degrades under variable lighting and dense canopy conditions. On the other hand, LiDAR provides higher range accuracy and robustness to illumination changes but at a higher cost and power consumption, which makes it not suitable for small-scale robots. Combining LiDAR geometric precision with stereo vision’s texture information improves row orientation accuracy by 15% to 20% compared to using either sensor alone [65,66]. Similarly, low-cost ultrasonic or infrared sensors are suitable for obstacle avoidance, but their short range and limited angular resolution reduce reliability in cluttered environments. Hence, sensor selection often involves a trade-off between accuracy and cost, emphasizing the importance of hybrid and multimodal sensing strategies in precision agriculture. Table 5 illustrates sensors used in the positioning of agriculture systems.

**Table 5 sensors-25-06844-t005:** The recent major upgrades of individual GNSS components and their impact on precision agriculture [67].

GNSS	Recent Upgrades	Impact on Precision Agriculture
GPS	GPS III satellites [68]	Improved signal strength
	L5 civil signal [69,70]	Increased resistance to multipath interference
GLONASS	GLONASS-K satellites [71,72]	Increased satellite availability and signal strength
Galileo	Full operational capability [73]	Global coverage and reliable signal reception
	High Accuracy Service [74]	Centimeter-level positioning accuracy
BeiDou	BeiDou-3 satellites [75,76]	Global coverage and reliable signal reception
	New signals (B1C, B2a, B2b) [77,78]	Improved positioning accuracy

In agricultural robotics, a wide array of sensors is employed, including force-torque, tactile, encoders, infrared, sonar, ultrasonic, gyroscopes, accelerometers, active beacons, laser range finders, and vision-based sensors such as color tracking, proximity, contact, pressure, and depth sensors [79]. Stereo cameras, which use multiple lenses and image sensors, are especially common for plant localization, allowing robots to accurately map the position of crops in real time [26,31,49,53,56,65,66,80,81]. Furthermore, object-detection AI models, such as YOLOv3 (You Only Look Once, Version 3), play a pivotal role in real-time object recognition. YOLOv3 utilizes deep convolutional neural networks (CNNs) to identify objects in videos, live streams, or still images, enabling agricultural robots to localize plants and other objects with high precision [66,82,83].

A central element of precision agriculture (PA) is the wireless sensor network (WSN), which consists of multiple wireless nodes that collect environmental data through various sensors. These nodes, which include a micro-controller, radio transceiver, sensors, and antenna, are connected to one another and transmit data to a central system for processing and analysis [84]. The ability to monitor soil conditions, crop health, and environmental variables in real time has become possible due to advancements in WSN technologies, leading to a reduction in the size and cost of sensors. This has made sensor deployment feasible in diverse agricultural applications, as illustrated in Table 6, which lists common sensors used for aiding precision agriculture [85].

Wireless sensor nodes are typically categorized into source nodes, which gather the data, and sink nodes, which aggregate and transmit the data to the central system. Sink nodes are more powerful, offering enhanced computational and processing capabilities compared to source nodes. However, choosing the right wireless node depends on several aspects, such as power, memory, size, data rate, and cost. Table 7 shows a comparison of various wireless nodes and their specifications, highlighting their suitability for agricultural sensing and localization applications. Among these, the MICA2 wireless node is particularly notable due to its numerous expansion connectors, making it an ideal choice for connecting to multiple sensors and supporting complex monitoring tasks in agriculture [85].

**Table 7 sensors-25-06844-t007:** Wireless nodes and their sensors [86].

WN1	MC2	Expansion Connector	Available Sensors	Data Rate
1	MICA2DOT	ATmega128L	GPS, Light, Humidity, Barometric pressure, Temperature, Accelerometer, Acoustic, RH	38.4 K Baud
2	Imote2	Marvell/XScalePXA271	Light, Temperature, Humidity, Accelerometer	250 Kbps
3	IRIS	ATmega128L	Light, Barometric pressure, RH, Acoustic, Acceleration/seismic, Magnetic and video	250 Kbps
4	MICAz	ATmega128L	Light, Humidity, Temperature, Barometric pressure, GPS, RH, Accelerometer, Acoustic, Video sensor, Sounder, Magnetometer, Microphone	250 Kbps
5	TelosB	TIMSP430	Light, Temperature, Humidity	250 Kbps
6	Cricket	ATmega128L	Accelerometer, Light, Temperature, Humidity, GPS, RH, Acoustic, Barometric pressure, Ultrasonic, Video sensor, Microphone, Magnetometer, Sounder	38.4 K Baud
7	MICA2	ATmega128L	Temperature, Light, Humidity, Accelerometer, GPS, Barometric pressure, RH, Acoustic, Sounder, Video, Magnetometer	38.4 K Baud

WN1: Wireless node, MC2: Micro-controller.

Moreover, the Robot Operating System (ROS) is extensively utilized in agricultural robotics for facilitating communication between hardware and software components. The ROS is an open-source framework that has significantly advanced robotics applications in agriculture. Researchers commonly use Python or C++ for ROS programming [31,48,49]. In agricultural robotics, the ROS is organized around core stacks such as perception, localization and mapping, planning, control, and navigation. Most implementations rely on ROS 1, which is more widely supported by open-source libraries and has a larger community for debugging [38]. The adoption of precision agricultural mobile robots is expected to grow based on the decreasing prices of sensors and the availability of open-source platforms. Figure 3 illustrates some of the basic localization and sensing components used in agricultural robots.

### 3.4. Path Planning and Navigation

Path planning is the process of calculating a robot’s continuous journey from an initial state to a goal state or configuration [46]. This method is based on a preexisting map of the surroundings stored in the robot’s memory. The state or configuration describes the robot’s potential position in the environment, and transitions between states are accomplished by particular actions [8]. Effective path planning is essential for robotic control and must satisfy several criteria, including collision avoidance, reachability, smooth movement, minimized travel time, optimal distance from obstacles, and minimal abrupt turns or movement constraints [87]. In agricultural applications, like fruit harvesting, path planning is influenced by the type of manipulator, end effector, and crop being harvested. Path planning becomes computationally intensive with manipulators that have many degrees of freedom (DOF), though efficiency improves significantly when the DOF is limited to the requirements of the task [81].

Path planning algorithms are employed across various applications, including autonomous vehicles, unmanned aerial vehicles, and mobile robots, to determine safe, efficient, collision-free, and cost-effective paths from a starting point to a destination [88]. Depending on the environment, there may be multiple viable paths—or none—connecting the start and target configurations. Additional optimization criteria, such as minimizing path length, are often introduced to achieve specific objectives.

Robot navigation pertains to the robot’s ability to determine its location within the environment and plan a route to a specified destination. This requires both a map of the environment and the capacity to interpret it. In open fields, robots often utilize GPS and cameras for navigation, employing path-tracking algorithms. In contrast, robots in greenhouses are typically guided by tracks; therefore, they need position-control algorithms instead of full navigation algorithms [38,39,48,53,54,55,56,57]. Some research has focused on motion planning for robotic arms without incorporating obstacle avoidance or traditional path-planning mechanisms [64,65,89]. Recent advancements have introduced sophisticated path-planning approaches in agriculture, such as leveraging convolutional neural networks (CNNs) [49,80] and navigating along predefined paths mapped in advance [38,48].

## 4. AI for Precision Agriculture

### 4.1. Available Datasets

#### 4.1.1. Multispectral Dataset for Tomato Disease Detection

In 2022, a multispectral dataset was presented by Georgantopoulos et al [90], intended for identifying plant diseases and pests in tomato crops. The dataset includes 314 multispectral images that combine three RGB channels with two Near-Infrared (NIR) channels (850 nm and 980 nm), recorded in real greenhouse settings using a MUSES9-MS-PL multispectral camera. It focuses on two major tomato diseases: *Tuta absoluta* (tomato leaf miner) and *Leveillula taurica* (powdery mildew). A visual representation of the dataset’s multispectral cube, highlighting the captured NIR and visible spectrum channels [90], is shown in Figure 4.

**Figure 4 sensors-25-06844-f004:**
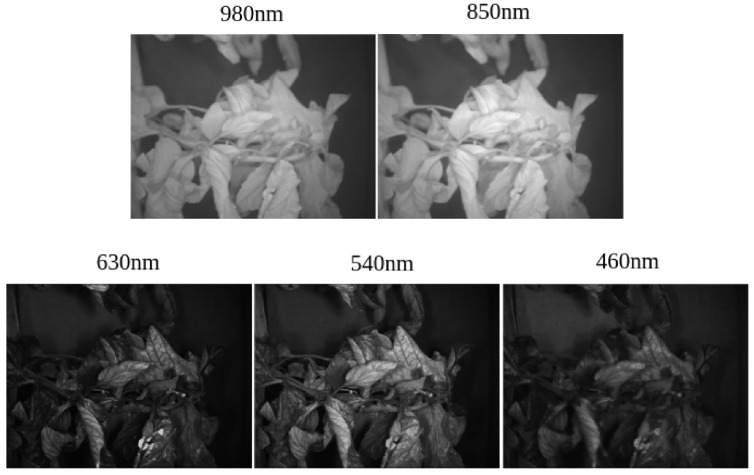
A multispectral cube of the dataset. **Top**: the NIR channels at 850 nm and 980 nm. **Bottom**: channels of the visible spectrum in the wavelength regions of red, green, and blue (see Table 8). (For interpretation of the references to color in this figure legend, the reader is referred to the web version of this article [90]).

**Table 8 sensors-25-06844-t008:** Wavelength regions supported by the MUSES9-MS-PL camera and those used in the current study [90].

Spectral Band	Lower Limit (nm)	Upper Limit (nm)	Current Study (nm)	FWHM * (nm)
Infrared	800	1000	980, 850	50
Red	600	700	630	40
Green	500	600	540	30
Blue	400	500	460	30
Ultraviolet	365	385	-	-

* FWHM: Full Width at Half Maximum.

The dataset was created in a greenhouse environment. The infected tomato plants were classified based on various stages of disease development. The images were then annotated with bounding boxes for lesions, and the levels of lesion stage progression were tagged to show the disease severity. The addition of NIR channels improves disease detection by offering more spectral data for separating backgrounds and locating lesions. This dataset is an important tool for training machine learning models aimed at automated disease identification. As reported by Georgantopoulos et al. [90], a Faster R-CNN baseline (with a ResNet-50 backbone, default anchor configuration, and 512 × 512 input resolution) was trained and evaluated on this dataset. Using an 80/10/10 train–validation–test split and an IoU threshold of 0.5, the model achieved a mean Average Precision (mAP) of 20.2 across the target classes, as detailed in their evaluation protocol. The open-access nature of the dataset promotes additional studies in multispectral imaging for precision agriculture.

#### 4.1.2. PlantVillage Dataset

As of 2024, the PlantVillage dataset [14,15] contains over 54,000 RGB images. These images are categorized into 38 distinct classes, each representing a specific plant disease, and are further divided into two main groups: healthy and infected (sick) crop leaves. Each original RGB image is accompanied by a grayscale version and a segmented version, providing additional modalities for model training and evaluation. The inclusion of these variants allows for the exploration of advanced preprocessing and feature extraction techniques, enhancing the versatility of the dataset. Additionally, the dataset is organized into training and validation sets in an 80/20 ratio while maintaining the original directory structure. The dataset contains a variety of species, as shown in Table 9 and Figure 5. Some sample images from the dataset can be seen in Figure 6 [91].

**Figure 6 sensors-25-06844-f006:**
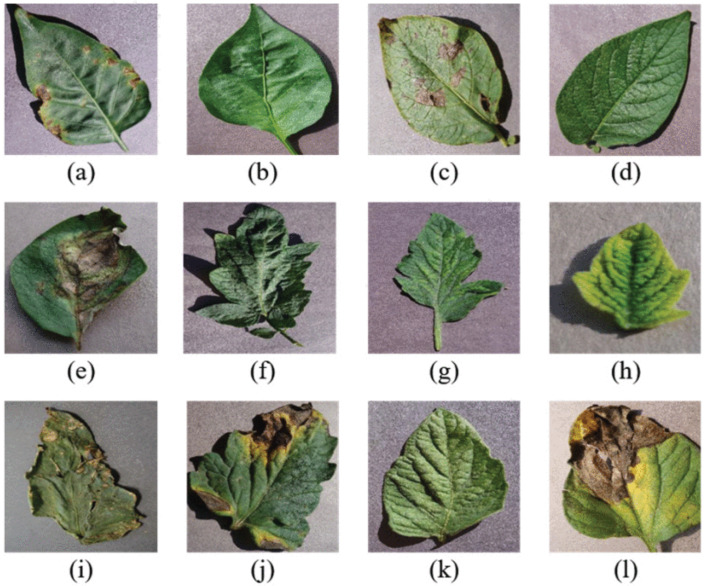
Some sample images from the PlantVillage dataset [92]. (**a**) Pepper bell—bacterial spot, (**b**) Pepper bell—healthy, (**c**) Potato—Early blight, (**d**) Potato—healthy, (**e**) Potato—late blight, (**f**) Tomato—target spot, (**g**) Tomato—tomato mosaic virus, (**h**) Tomato—tomato yellow leaf curl virus, (**i**) Tomato—bacterial spot, (**j**) Tomato—early blight, (**k**) Tomato—healthy, (**l**) Tomato—late blight.

**Table 9 sensors-25-06844-t009:** Summary of the PlantVillage dataset [93].

Species	Diseases	Healthy Categories	Number of Images
Apple	3	1	33,172
Blueberry	0	1	1502
Cherry	1	1	1906
Corn	3	1	3852
Grape	3	1	4063
Orange	1	0	5507
Peach	1	1	2657
Bell Pepper	1	1	2475
Potato	2	1	2152
Raspberry	0	1	371
Soybean	0	1	5090
Squash	2	0	1835
Strawberry	1	1	1565
Tomato	9	1	18,162

#### 4.1.3. PlantDoc Dataset

In 2020, the PlantDoc dataset [16] was introduced to overcome the limitations of the PlantVillage dataset, which primarily features images captured under controlled conditions. While the controlled environment of the PlantVillage dataset ensures consistency, it limits the dataset’s effectiveness for real-world disease detection, where plant images often include multiple leaves amidst diverse background conditions and varying lighting. In contrast, the PlantDoc dataset was specifically designed to address these challenges, consisting of 2598 images spanning 13 plant species and 27 classes, of which 17 represent diseases and 10 represent healthy leaves. Notably, PlantDoc is the first dataset to provide data captured in non-controlled environments, enhancing its utility for detecting plant diseases in realistic agricultural settings.

The dataset not only benchmarks curated data for disease detection but also showcases its ability to navigate the complexities of natural settings. For instance, Figure 7 illustrates the statistical breakdown of leaf diseases within the PlantDoc dataset, providing a comprehensive overview of its class distribution. Additionally, Figure 8 compares sample images captured under laboratory and field conditions, highlighting the dataset’s capacity to reflect real-world scenarios, such as diverse backgrounds and fluctuating lighting. These features make PlantDoc an invaluable resource for practical applications in agricultural disease detection.

#### 4.1.4. FieldPlant Dataset

In 2023, the FieldPlant dataset [17] was introduced to address the limitations of the PlantVillage [14,15] and PlantDoc [16] datasets. The PlantVillage dataset that is almost always used in plant disease detection research is mostly composed of leaf images taken under controlled conditions. While this maintains consistency, it limits the dataset’s applicability in real-world scenarios. The PlantDoc dataset, on the other hand, combines both field-captured and web-sourced images collected from diverse environments to reflect real-world variability in lighting, background, and leaf conditions [16]. While some images were gathered from online sources, the original PlantDoc paper emphasizes that many were contributed directly by agricultural researchers and practitioners to better represent natural farm settings.

The FieldPlant dataset doesn’t have these limitation as it consists of photos captured directly from farms. That dataset consists of 5170 annotated images of leaves collected from Cameroonian plantations, with a significant focus on three tropical crops: maize, cassava, and tomatoes. These images were precisely annotated using the Roboflow platform and classified into 27 disease classes by a group of plant pathologists. In total, the dataset has 8629 different leaf annotations, creating a robust source for training and evaluating machine learning models.

Images were captured by smartphone cameras with a resolution of 4608×3456 pixels (4:3 aspect ration). Figure 9 shows the areas in Cameroon where the FieldPlant dataset images where taken, it also shows the environmental diversity and conditions of areas where the images were taken. Figure 10 shows the statistics of the dataset class distribution. Sample images include a corn single-leaf image depicted in Figure 11 and a tomato multiple-leaf image shown in Figure 12, showing the that the dataset is relevant in both single leaf and multi-leaf analysis. All of this makes the FieldPlant dataset perfect for advancing plant disease detection in tropical crop systems.

### 4.2. Deep Learning Model

Deep learning (DL) algorithms utilize deep neural networks with multiple hidden layers to imitate the human cortex operations [94]. Convolutional neural networks (CNNs), which are deep learning algorithms, are best for large datasets and extracting very complex features from 2D images [95]. Deep learning architectures have been instrumental in automating plant disease detection, weed detection, and crop classification. Their design characteristics influence how they perform in agricultural imaging.

#### 4.2.1. VGG16

In 2015, VGG-16 [18], a deep CNN, was used for small datasets such as CIFAR-10, showing its ability to handle small-scale image classification tasks. The research shows how models that are built to handle large datasets can be modified to also handle smaller datasets. It stressed the different challenges that small datasets pose, like overfitting and feature vanishing, while using the power of the VGG-16 architecture.

The input size was changed to adapt the the CIFAR-10 dataset, which has images of 32 × 32. VGG-16 consists of 13 convolution layers grouped into five sets, then 3 fully connected layers, but in this study, the architecture was modified. The two 4096 -dimensional fully connected layers were shrunk to one 100-dimensional layer, which helped decrease the number of parameters and avoid overfitting. The filter size of 3 × 3, with a stride of 1 and a pooling region of 2 × 2, was applied without overlap throughout the network.

To improve the performance, batch normalization was added before each non-linearity layer, as it helps remove the problem of internal covariate shift, which leads to slow convergence. This stabilized training, resulting in faster convergence, lower error rates, and reduced overfitting. Another modification was the use of strong dropout settings in both the fully connected layers and the convolution layers. The dropout rate was different among different convolution layers, as deeper layers had higher dropout rates. Dropout and batch normalization, when used together, are effective in preventing overfitting on small datasets.

Various activation functions were explored, but no significant performance impacts were observed, The authors suggested that fine-tuning negative slope settings for leaky ReLU could lead to further optimization.

When it comes to acceleration techniques, the authors tried replacing the current 3 × 3 convolution layer with 5 × 5 layers. This modification decreased the computational time, but at the same time increased the error rate, highlighting the trade-off between speed and accuracy.

The study successfully shows that VGG-16 can be modified to perform well with small datasets like CIFAR-10. Modifications like batch normalization and strong dropout settings can help avoid challenges like overfitting and feature vanishing. See Figure 13 for the full architecture.

**Figure 13 sensors-25-06844-f013:**
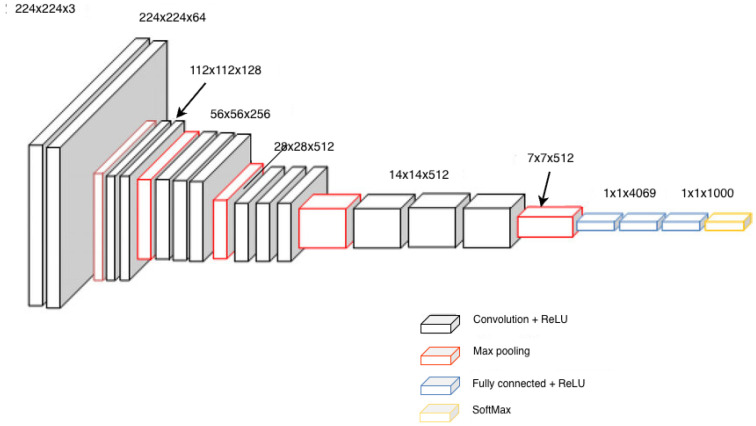
Architecture of VGGNet [96].

VGG-16 demonstrated its effectiveness when it achieved a classification accuracy of 12.75% when transferring from PlantVillage (100%) to PlantDoc (100%), 40.3% when evaluated on PlantDoc (80% training and 20% testing), and 80.54% on FieldPlant (80% training and 20% testing). These results underscore the model’s capacity to handle different dataset complexities, performing particularly well in datasets that simulate real-world conditions, such as FieldPlant, as shown in Table 10.

VGG16 helps advance precision agriculture for disease classification tasks when high accuracy is needed and the image complexity is moderate. VGG16 has many convolution layers, which makes it good at capturing fine spatial details. For example, it achieved 95.62% accuracy in detecting rice plant diseases under transfer learning [97]. As previously mentioned, VGG16 has many convolution layers, which leads to slower inference and high memory usage, which makes deployment on edge devices more challenging [98]. Due to its deep structure and large number of parameters, it performs best on high quality datasets like PlantVillage but struggles with generalization to real-field conditions.

#### 4.2.2. InceptionV3

In 2017, InceptionV3 [20], a deep and efficient convolutional neural network, was introduced specifically for image classification tasks. The architecture builds upon earlier Inception models, integrating advanced design principles and optimizations to enhance computational efficiency and predictive performance across diverse applications. Notable innovations include the factorization of convolutions, where larger spatial filters (e.g., 5 × 5 or 7 × 7) are replaced by sequences of smaller 3 × 3 convolutions, reducing computational costs without compromising the network’s representational power. Additionally, asymmetric convolutions replace traditional n × n convolutions with a 1 × n convolution followed by an n × 1 convolution, significantly lowering computational expense, particularly for medium grid sizes (e.g., 12 × 12 to 20 × 20).

Auxiliary classifiers, initially introduced in earlier Inception models to address vanishing gradient problems, serve as regularizers in InceptionV3, improving overall network performance during training. Furthermore, efficient grid size-reduction techniques replace traditional methods, such as pooling followed by convolution, with parallel stride-2 blocks: one employing a pooling layer and the other a convolutional layer. By concatenating their outputs, the network reduces grid size without introducing representational bottlenecks, effectively balancing computational efficiency and performance.

The architecture of InceptionV3 shown in Figure 14 represents a significant improvement over previous benchmarks. It replaces 7 × 7 convolutions with three sequential 3 × 3 convolutions and includes three traditional Inception modules operating on a 35 × 35 grid with 288 filters each. Using grid reduction techniques, it transitions to a 17 × 17 grid with 768 filters, followed by five factorized inception modules that reduce the dimensions further, to an 8 × 8 × 1280 grid. At the coarsest 8 × 8 level, the architecture employs two additional Inception modules, culminating in a concatenated output filter bank size of 2048. Despite being 42 layers deep, InceptionV3 maintains a computational cost only 2.5 times that of GoogLeNet [99] while being significantly more efficient than VGGNet [18].

When applied to plant disease datasets, InceptionV3 demonstrated its versatility and effectiveness. It achieved a classification accuracy of 14.25% when transferring from PlantVillage (100%) to PlantDoc (100%), 51.27% when evaluated on PlantDoc (80% training and 20% testing), and 82.54% on FieldPlant (80% training and 20% testing) [17]. These results underscore the model’s ability to adapt to varying dataset complexities, excelling in datasets with realistic field conditions, such as FieldPlant, as shown in Table 11.

InceptionV3 is more suitable for sophisticated feature extraction and better handling of scale variation and different feature sizes. Its architecture helps capture both broad context and minute details [100]. In recent cassava disease classification tasks, as well as in edge computing settings for thermal imaging, it was among the models that achieved strong trade-offs between accuracy and inference performance when deployed on edge devices [98]. InceptionV3 is still more computationally expensive than MobileNet, but less so than very large and deep models. The multi-scale convolutional filters InceptionV3 has allow it to capture complex plant structure under different natural light conditions, see Figure 14.

**Figure 14 sensors-25-06844-f014:**
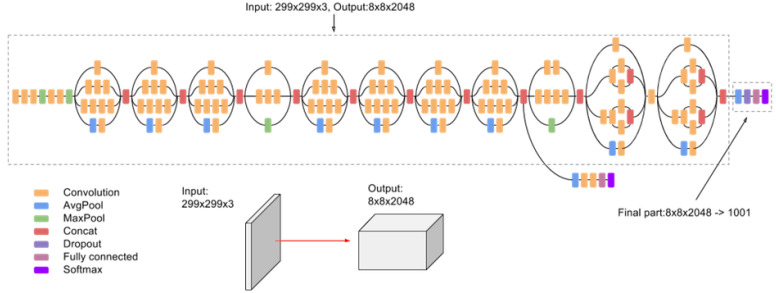
InceptionV3 architecture [101].

#### 4.2.3. MobileNet

In 2017, MobileNet [19], a lightweight and efficient deep learning model, made its debut specifically for image classification tasks on resource-constrained devices.

The input size for the baseline MobileNet model is 224×224 pixels. However, the resolution can be adjusted using the resolution multiplier (ρ) hyperparameter, enabling reduced-computation MobileNet variants with input resolutions of 192×192, 160×160, or 128×128. This adaptability allows MobileNet to be tailored to various computational constraints and application requirements.

The MobileNet architecture shown in Figure 15 is mainly based on depthwise separable convolutions, a factorization technique, and model size. The convolution operation is separated into two steps, depthwise convolutions and pointwise convolution, which uses 1×1 convolution to combine these filters. This factorization enables a substantial reduction in computation, achieving an efficiency improvement of 8 to 9 times compared to standard convolutions, with minimal loss in accuracy.

**Figure 15 sensors-25-06844-f015:**
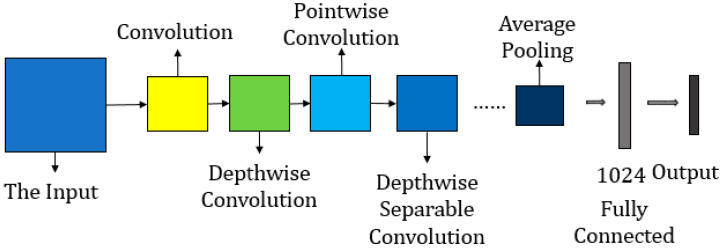
MobileNet architecture [102].

The architecture comprises 28 layers, treating depthwise and pointwise convolutions as separate layers. The first layer performs a full convolution, while all subsequent layers (except the final fully connected layer) are followed by batch normalization and ReLU nonlinearity. Down-sampling is achieved through strided depthwise convolutions and the first convolutional layer. An average pooling layer reduces spatial resolution before the fully connected layer, further enhancing computational efficiency.

MobileNet incorporates additional enhancements over traditional architectures. Depthwise separable convolutions leverage 1×1 convolutions to optimize general matrix multiplication (GEMM) functions, further improving efficiency. The introduction of width multiplier (α) and resolution multiplier (ρ) hyperparameters offers flexibility, allowing users to create smaller and faster MobileNet versions without designing a new architecture. This adaptability makes MobileNet highly suitable for a range of tasks, including image classification, object detection, and fine-grained recognition, while maintaining a significantly smaller size and computational cost compared to models such as VGG16 [18], GoogleNet [99], and AlexNet [103].

When applied to plant disease datasets, MobileNet demonstrated notable performance. It achieved a classification accuracy of 16.75% when transferring from PlantVillage (100%) to PlantDoc (100%), 60.14% on PlantDoc (80% training and 20% testing), and 82.9% on FieldPlant (80% training and 20% testing). These results illustrate MobileNet’s ability to balance computational efficiency and performance, particularly in datasets with realistic field conditions, such as FieldPlant, as shown in Table 12.

### 4.3. Applied Analysis of AI Models in Agriculture

Deep learning models have demonstrated wide applicability across agricultural domains beyond disease detection. Table 13 shows some of those uses.

**Table 13 sensors-25-06844-t013:** Comparison of deep learning models in precision agriculture.

Model	Strengths	Applications	Cited Work
**VGG16**	High accuracy and robust feature extraction; strong transfer learning capabilities.	Disease classification; soil fertility mapping.	[15,104]
**InceptionV3**	Multi-scale feature extraction; good accuracy–efficiency trade-off.	Weed detection; yield prediction.	[105,106]
**MobileNet**	Lightweight and fast, suitable for edge devices with limited compute power.	Pest detection; real-time field monitoring.	[107,108]

## 5. AI and Thermal Imaging in Precision Agriculture

Thermal imaging serves as an essential tool in scientific research due to its ability to provide unique thermal information that is not available through standard RGB images [109]. Unlike RGB images, where pixel intensity represents color values (red, green, or blue), thermal images encode temperature as pixel intensity, offering a detailed thermal perspective of objects [110,111,112]. This capability is particularly valuable in agriculture, where leaf temperature is a standard indicator of plant water stress. By analyzing thermal data, it becomes possible to distinguish between stressed and unstressed plants, making thermal imaging a vital resource in monitoring plant health. Table 14 shows how thermal imaging outperforms RGB, which is a traditional monitoring method for precision agriculture [113,114].

Thermal images, paired with their accompanying colorbars, deliver abundant and precise thermal details that are crucial for accurate analysis; see Figure 16.

In contrast, RGB images provide visual representations but lack the temperature-based insights that thermal imaging offers; see Figure 17. These unique advantages also make thermal imaging indispensable for broader applications, such as object detection [115], object classification [116], and scene reconstruction [117].

**Table 14 sensors-25-06844-t014:** Comparison of thermal imaging (TI) and RGB imaging in precision agriculture.

Aspect	Thermal Imaging (TI)	RGB Imaging	Cited Work
**Water Stress Detection (Wheat)**	Achieved **98.4% accuracy** with ResNet50 under varied irrigation.	Reached **96.9%**, less reliable under light variation.	[118]
**Water Stress Detection (Okra)**	**84–88% accuracy** using TI + DL models.	**74–79%** using RGB data only.	[119]
**Pest Infestation (Maize)**	Detected **+3.3 °C canopy rise** in FAW-infested crops **before visible symptoms**.	Detected only **post-damage discoloration**.	[120]
**Stored Grain Infestation**	Identified internal heat from insect activity **before surface damage**.	Failed to detect internal infestations.	[121]

### 5.1. The Role of AI and Thermal Imaging in Precision Agriculture

As of 2024, the integration of artificial intelligence (AI) and machine learning (ML) into agriculture has revolutionized the detection and management of plant pests and diseases [122,123]. Mirzaev and Kiourt et al. highlighted the transformative role of thermal imaging (TI) in precision agriculture, particularly when paired with advanced AI models. TI captures subtle variations in plant temperature that are imperceptible to the human eye, enabling the identification of early-stage infestations and infections. For example, AI thermal analysis is being used to detect aphid infestation in crops where pests drain sap and endanger crop health [124]. Also, spider mites in crops like cotton are recognized by temperature changes that are caused by their feeding activity [125]. These early detections help significantly reduce crop losses [126]. AI alone, or even when powered by traditional perception methods like RGB imaging, is limited and cannot capture the certain diseases that do not show any physical symptoms that thermal imaging can capture. Additionally, thermal imaging can detect certain diseases earlier.

AI, when used alone or combined with traditional perception methods such as RGB imaging, is limited and cannot detect certain diseases that show no visible symptoms but can be identified through thermal imaging.

Thermal imaging also helps in detecting fungal diseases, which can grow in crops like wheat, as small variations in temperature even before visual symptoms appear. AI and deep learning (DL) algorithms process the data to find the affected areas precisely, giving farmers time to take preventive measures. Figure 18 is an example that shows how thermal imaging works, by converting a normal RGB image to a thermal image.

**Figure 18 sensors-25-06844-f018:**
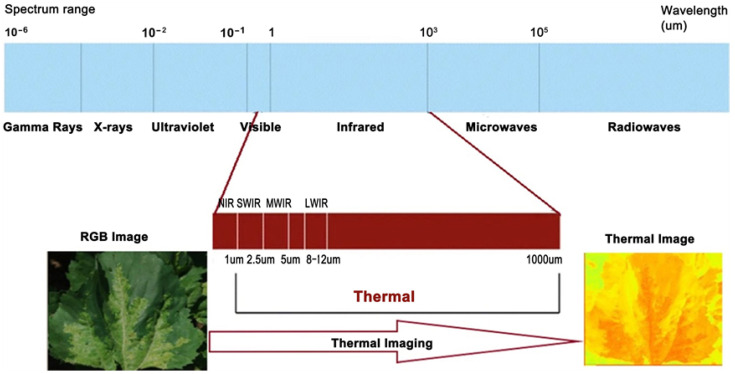
Description of thermal imaging conversion [127].

AI advances the abilities of thermal imaging by automating the analysis and classification of data, delivering solutions specific to different crop and pest types. Convolutional neural networks have shown great accuracy in understanding thermal data for pest and disease identification.

AI-driven thermal imaging systems also help in optimizing irrigation by detecting water stress in crops, assessing plant growth stages, and predicting harvest readiness.

Together, AI, and TI represent a paradigm shift in agricultural diagnostics. They address the increasing challenges of pest and disease control in modern farming and also advance precision agriculture by improving efficiency and promoting environmental sustainability.

### 5.2. AI and Thermal Imaging in Post-Harvest Fruit Quality Assessment and Pest Detection

Furthermore, Pugazhendi et al. emphasized the transformative role of artificial intelligence (AI) in conjunction with thermal imaging (TI) for fruit quality assessment and pest detection. Thermal imaging, as a non-destructive and non-invasive tool, leverages the unique thermal signatures of fruits affected by pests or diseases, which result from alterations in their metabolic activity. By detecting subtle temperature variations, thermal cameras generate precise images that aid in the early identification of infested or infected produce. For instance, TI has been employed to monitor the surface temperatures of apples stored in plastic and cardboard containers [128], achieving exceptionally low root mean square error (RMSE) values of 0.410 °C and 0.086 °C, respectively, when tested on apple batches stored under controlled ambient conditions (22 ± 1 °C) in plastic and cardboard containers. This demonstrates TI’s effectiveness in maintaining optimal storage conditions and preventing post-harvest losses [129].

Further applications include optimizing post-harvest treatments, as shown in a study on Opuntia ficus-indica (cactus pear) [130]. The experiment involved brief cauterization of harvested cladodes at 200 °C for a few seconds to seal wounds and reduce microbial infection, significantly extending shelf life. This controlled thermal treatment highlights TI’s potential for guiding post-harvest process optimization in a safe and species-specific context. Additionally, TI has enhanced fruit detection within orange canopies by combining thermocouples with infrared cameras. A novel fusion of thermal and visible imaging, using fuzzy logic, outperformed traditional Local Pattern Texture (LPT)-based detection methods on an orange canopy dataset, achieving a detection accuracy of 94.3% compared to 88.6% for LPT. This quantitative improvement underscores the effectiveness of multimodal imaging for robust fruit detection and identification [129].

Incorporating AI into thermal imaging workflows expands its utility even further. Machine learning (ML) algorithms analyze large spectral datasets to identify patterns and attributes indicative of fruit quality [131]. Computer vision techniques use captured images to assess quality and detect surface defects like bruises and decay. However, both spectral imaging and computer vision face challenges such as high equipment costs, environmental sensitivity, and computational demands, which can hinder real-time applications. In contrast, TI offers distinct advantages, including its ability to provide consistent and objective data about temperature, moisture content, and other critical fruit characteristics [129].

Despite these advancements, Pugazhendi et al. note the need for further research to develop standardized protocols and algorithms for precise pest and disease detection using TI. Validating thermal signatures, optimizing imaging parameters, and integrating additional detection methods remain critical areas for investigation. Nonetheless, TI, when coupled with AI, holds immense promise as a rapid and reliable method to reduce post-harvest losses, ensure produce quality, and revolutionize agricultural practices [129].

An illustrative table, Table 15, accompanies this discussion, offering insights into the technology powering these applications.

## 6. Synthesis and Implications

Artificial intelligence integration with thermal imaging for precision agriculture has been advancing at a radical rate to attain high efficiency in pest and disease detection, crop monitoring, and post-harvest management. As shown in the discussion, AI models, especially deep learning algorithms such as CNNs, have shown exceptional performance in plant health classification and identifying subtle plant stress signals.

### 6.1. Challenges and Limitations

Despite the promising results, several challenges are yet to be overcome in the deployment of AI and TI systems in large-scale agricultural settings. The high cost of thermal cameras and the requirement for specialized equipment continue to hinder their widespread adoption. Besides, environmental factors such as temperature and humidity may affect the accuracy of the thermal imaging system. Such challenges can be minimized through further advances in technology and the development of cost-effective solutions.

Furthermore, the computational requirements for deep learning models, particularly in real-time applications, require high processing power, which is not always feasible in remote agricultural settings. More research on the efficiency of algorithms and hardware solutions is needed to overcome these limitations and make AI and TI technologies more accessible to farmers. The domain gap problem is one example—despite high accuracy in controlled datasets, some deep learning models underperform in real-life scenarios due to domain shifts caused by lighting, background, and occlusion differences. For instance, models trained on PlantVillage achieve 12% to 16% lower accuracy when tested on PlantDoc, stressing the need for domain adaption and data augmentation to enhance robustness [15,104].

### 6.2. Future Work

As AI and thermal imaging become increasingly sophisticated, so too will their uses in precision agriculture. Integration of AI models with other sensing technologies, such as multispectral and hyperspectral imaging, is the key for advancement in retrieving comprehensive data regarding plant health in future studies. Furthermore, establishing a standardized protocol for the validation of thermal signatures and optimization of parameters will also be critical in order to enhance the reliability of the system.

Most important in this development pipeline, though, is scalability for these AI-powered systems. The current models have tended to succeed in small-scale, heavily controlled environments, and their extension into larger and more varied farming landscapes will require further adaptation. By focusing on these areas, AI and TI systems can be further honed to be more reliable, economical, and impactful transformers of modern agriculture.

### 6.3. Conclusions

The integration of artificial intelligence, robotics, and thermal imaging forms a cohesive ecosystem rather than three separate technologies. AI algorithms help robots interpret complex sensory data from thermal and visual modalities, allowing autonomous systems to make decisions for tasks as disease detection, targeted irrigation, and precision spraying. Thermal imaging specifically boosts robotic perception by detecting early physiological stress responses that are invisible to traditional RGB sensors, providing vital inputs for AI models to analyze plant health. When integrated with robotic platforms, theses AI-thermal systems create closed-loop feedback mechanisms in which action, reasoning, and perception work together seamlessly to increase crop productivity.

## Figures and Tables

**Figure 1 sensors-25-06844-f001:**
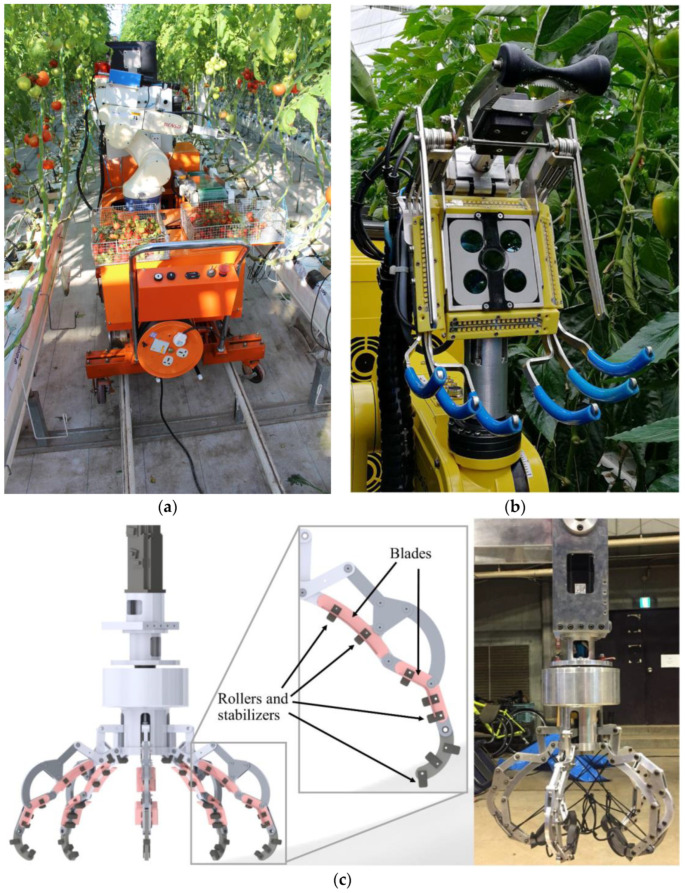
Robots with grasping and cutting concepts: (**a**) harvesting system for cherry tomato [26]; (**b**) sweet pepper harvesting robot [30]; (**c**) pumpkin harvesting robotic end effector [39].

**Figure 2 sensors-25-06844-f002:**
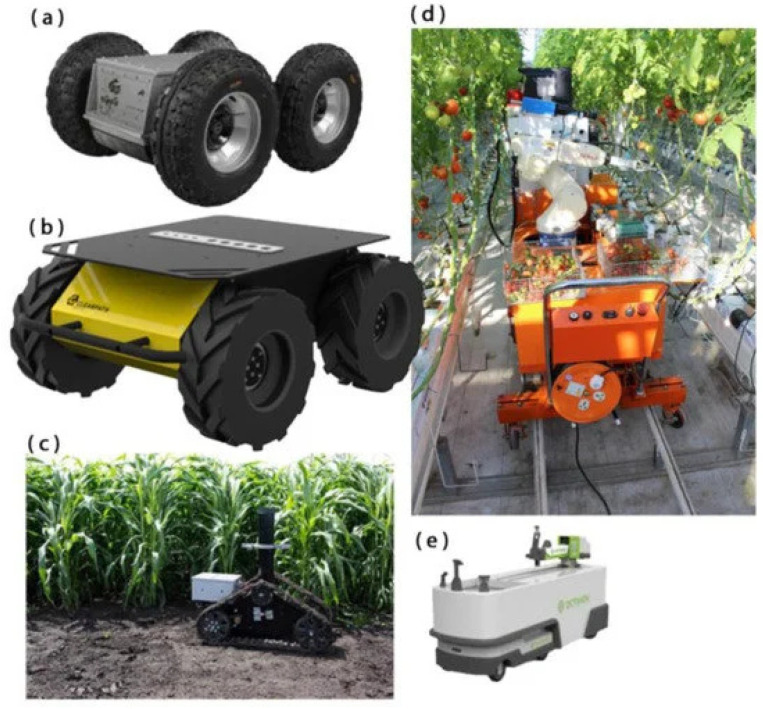
Robot mobile platform types: (**a**) railed platform [26]; (**b**) tracked platform (TERRA_MEPP) [57]; (**c**) independent steering devices (Octinion) [43]; (**d**) four-wheel platform (Segway) [49]; (**e**) four-wheel platform (Husky A200) [50].

**Figure 3 sensors-25-06844-f003:**
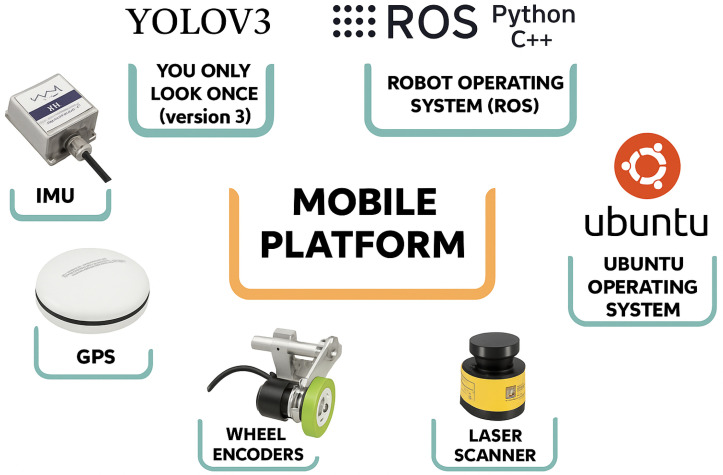
Localization and sensing components in agricultural robot systems [60].

**Figure 5 sensors-25-06844-f005:**
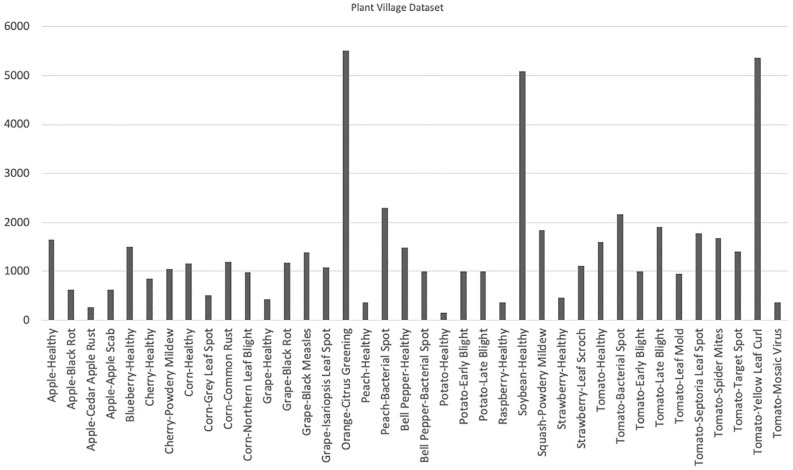
Statistics of PlantVillage dataset. Source: [17].

**Figure 7 sensors-25-06844-f007:**
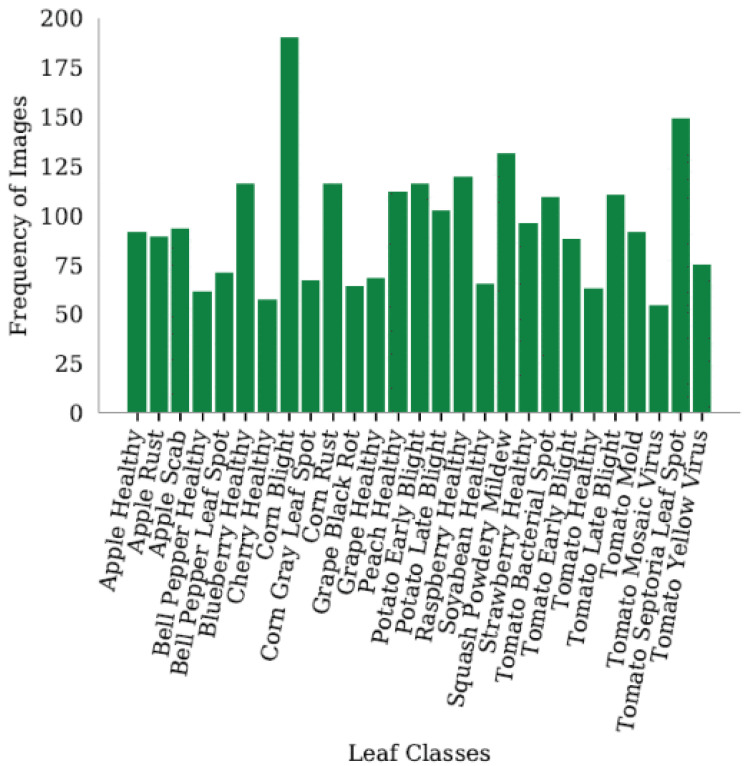
Statistics of PlantDoc dataset leaves with diseases [17].

**Figure 8 sensors-25-06844-f008:**
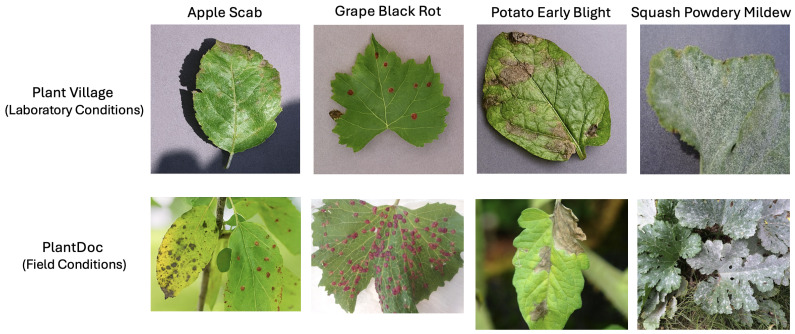
Some plant disease images under laboratory and field conditions [17].

**Figure 9 sensors-25-06844-f009:**
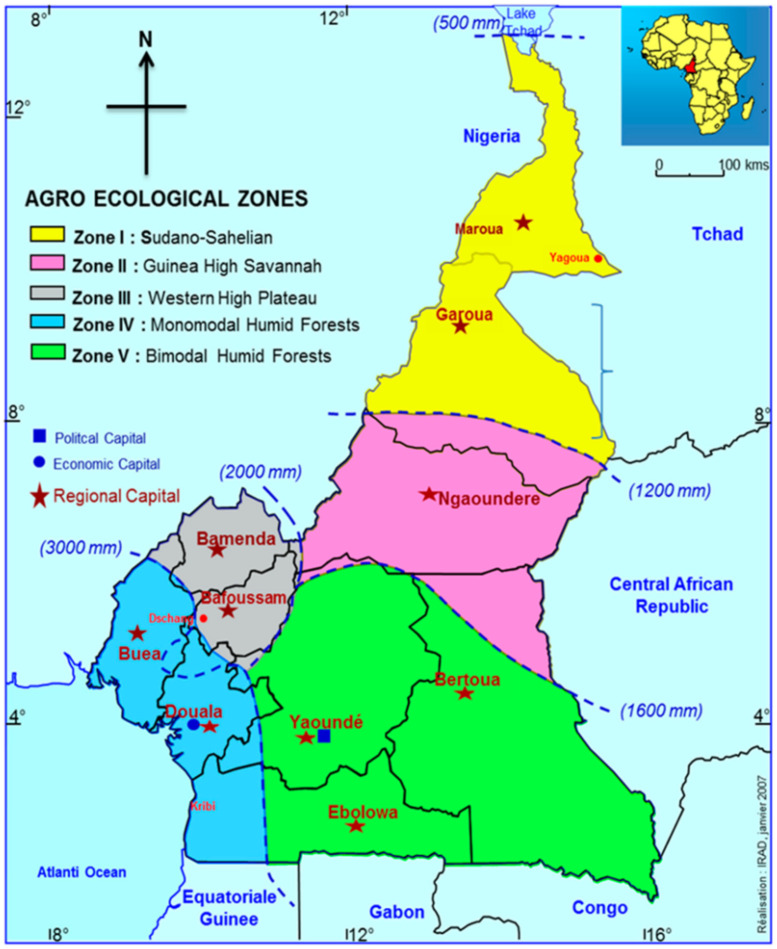
Agro -ecological zones of Cameroon [17].

**Figure 10 sensors-25-06844-f010:**
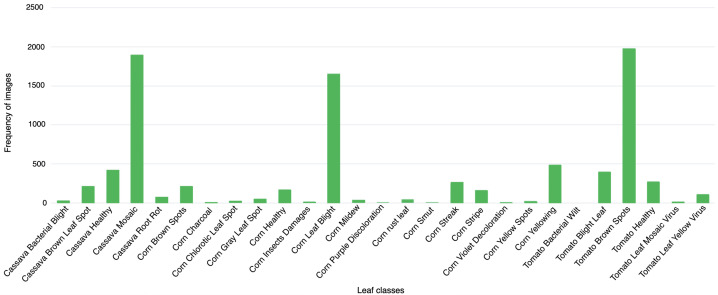
Statistics of Fieldplant dataset leaves with diseases [17].

**Figure 11 sensors-25-06844-f011:**
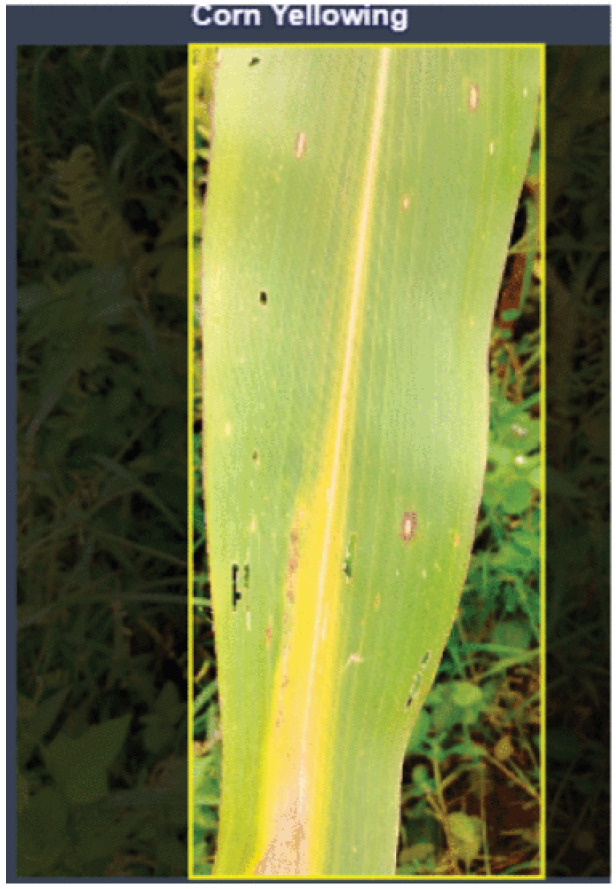
Corn single leaf image [17].

**Figure 12 sensors-25-06844-f012:**
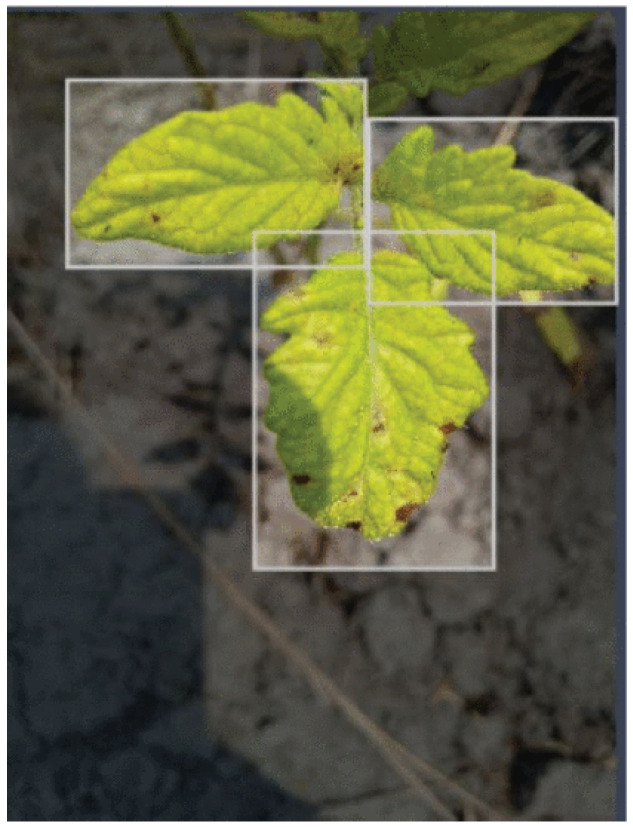
Tomato multiple leaves image [17].

**Figure 16 sensors-25-06844-f016:**
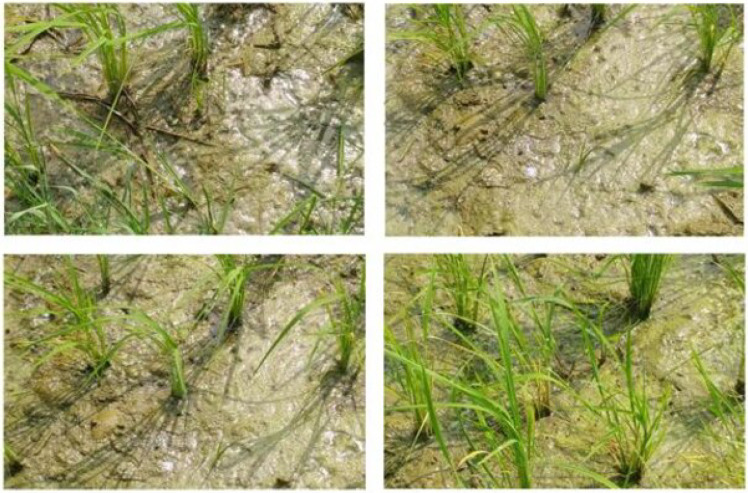
Sample of RGB images [113].

**Figure 17 sensors-25-06844-f017:**
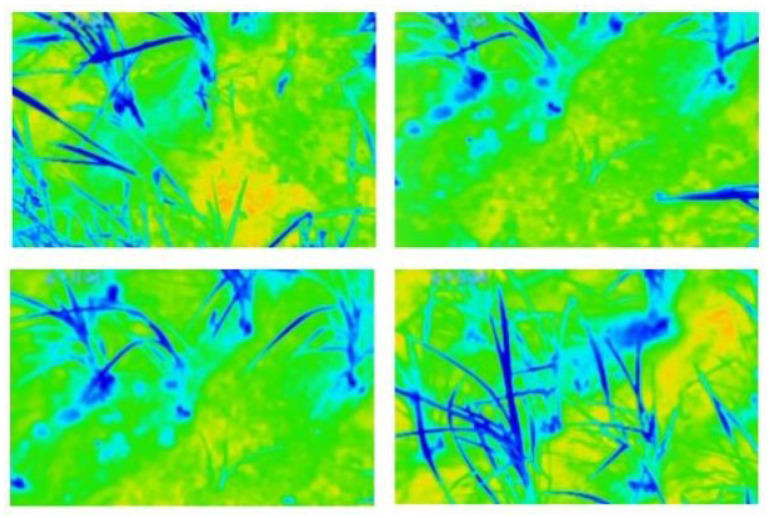
Sample of IR images [113].

**Table 1 sensors-25-06844-t001:** Performance of different crops in robotic harvesting.

Crop	Success Rate (%)	Damage Rate (%)	Speed (s/Cycle)	Cited Work
Sweet Pepper	61 (modified crop) and 18 (unmodified crop)	N/A	24	[30]
Cherry Tomato	83	N/A	8	[26]
Pumpkin	79	21	N/A	[39]

**Table 6 sensors-25-06844-t006:** Main sensors used in precision agriculture.

Sensor Type	Examples	Parameters	Advantages
Soil	Soil moisture, pH, EC sensors	Monitoring soil conditions (moisture, salinity)	Real-time, easy to deploy
Weather	Anemometers, Rain gauges	Monitoring local weather (wind, rain, temperature)	Optimize irrigation, pesticide use
Optical	NDVI, RGB cameras	Measuring plant health, canopy cover	Non-invasive, covers large area
Remote Sensing	LiDAR, Multispectral, Thermal, Satellite cameras	Topography mapping, vegetation analysis, crop stress detection	High accuracy, covers large area
Proximity	Ultrasonic, Capacitive, Inductive sensors	Estimating plant height, density, detecting growth stages	Affordable, simple to deploy
Nutrient	NPK sensors	Measuring soil nutrient levels	Direct nutrient management

**Table 10 sensors-25-06844-t010:** VGG-16 classification accuracy on different plant disease datasets [17].

CNN Model	Training (%)	Testing (%)	Accuracy (%)
VGG-16	PV (100%)	PD (100%)	12.75
VGG-16	PD (80%)	PD (20%)	40.3
VGG-16	FD (80%)	FD (20%)	80.54

Note: PV, PD, and FP represent PlantVillage, PlantDoc, and FieldPlant datasets, respectively.

**Table 11 sensors-25-06844-t011:** InceptionV3 classification accuracy on different plant disease datasets [17].

CNN Model	Training (%)	Testing (%)	Accuracy (%)
InceptionV3	PV (100%)	PD (100%)	14.25
InceptionV3	PD (80%)	PD (20%)	51.27
InceptionV3	FD (80%)	FD (20%)	82.54

Note: PV, PD, and FP represent PlantVillage, PlantDoc, and FieldPlant datasets, respectively.

**Table 12 sensors-25-06844-t012:** MobileNet classification accuracy on different plant disease datasets [17].

CNN Model	Training (%)	Testing (%)	Accuracy (%)
MobileNet	PV (100%)	PD (100%)	16.75
MobileNet	PD (80%)	PD (20%)	60.14
MobileNet	FD (80%)	FD (20%)	82.9

Note: PV, PD, and FP represent PlantVillage, PlantDoc, and FieldPlant datasets, respectively.

**Table 15 sensors-25-06844-t015:** Comparison of thermal cameras [129].

Model	Type	Sens. (mK)	Res.	Range (°C)	Price ($)
FLIR ONE edge Pro	Smart	70	160 × 120	−20 to 400	550
FLIR 5C	Handheld	70	160 × 120	−20 to 400	799
RevealPRO	Handheld	70	320 × 240	−40 to 330	699
Flir One Gen 3	Smart	150	80 × 60	−20 to 120	229
GTC400C	Handheld	50	320 × 240	−40 to 400	966
Compact Pro	Smart	70	160 × 120	−40 to 330	499
FLIR TG267	Handheld	70	320 × 240	−25 to 380	549
ShotPro	Handheld	70	160 × 120	−40 to 330	699
Scout TK	Handheld	50	160 × 120	−20 to 40	649
CAT S62 Pro	Smart	150	160 × 120	−20 to 400	579
E8-XT	Handheld	50	320 × 240	−20 to 550	619
IR202	Smart	150	80 × 60	−40 to 400	140

Note: Sens. and Res. represent Sensitivity and Resolution, respectively.

## Data Availability

The original contributions presented in this study are included in the article. Further inquiries can be directed to the corresponding author.

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
