# Peer review of "Fusion of Robotics, AI, and Thermal Imaging Technologies for Intelligent Precision Agriculture Systems"

_sensors, 2025, doi:10.3390/s25226844_

Round 1

Reviewer 1 Report

Comments and Suggestions for Authors

This manuscript presented a timely and comprehensive review on the integration of advanced technologies—robotics, artificial intelligence (AI), and thermal imaging (TI)—for precision agriculture. It was well-structured and provided a valuable synthesis of a wide range of literature, particularly in detailing robotic components and available datasets for plant disease detection. The section on AI and thermal imaging was a notable strength, highlighting a promising research direction. However, the manuscript in its current form would benefit from significant revisions to enhance its analytical depth, clarity, and scholarly impact. The primary concerns are a need for deeper critical analysis, stronger integration between the three core technological themes(Robotics, AI and TI), and improved presentation of figures and tables. Detailed comments are as follows:

  1. The current title accurately listed the three key technologies but presented them as parallel components. The manuscript's potential should be exploring their synergy. I suggest refining the title to better reflect this integrative focus. Correspondingly, the introduction and conclusion should more explicitly frame the review around how these technologies complement each other.
  2. There were 7 keywords in this manuscript, should be reduced.
  3. Sections 3.1 and 3.2 provided a good descriptive overview of robotic end-effectors and platforms. However, a critical comparative analysis should be provided. A summary table comparing different mechanisms (e.g., success rates, damage rates, speed) and platforms (e.g., cost, terrain adaptability, payload) with both advantages and disadvantages should be listed.
  4. While Section 6 effectively established the value of AI-powered thermal imaging, the rationale for this specific combination needed strengthening. The manuscript should more clearly articulate why thermal imaging is a uniquely powerful data source for AI models in agriculture compared to other sensing modalities, thereby justifying its focus.
  5. Figure 5, figure 10, figure 15 and figure 18 were not clear.
  6. The order of Figures 13 and 14 should be reversed.
  7. For Section 7, figure 16-18 were not cited in the main text, and the font size was wrong. The conclusion is too generic. It should synthesize the key findings for each technology area and, most importantly, provide a clear, forward-looking statement on the most promising pathways for integrating robotics, AI, and thermal imaging.

Author Response

Reviewer 1
This manuscript presented a timely and comprehensive review on the integration of advanced technologies—robotics, artificial intelligence (AI), and thermal imaging (TI)—for precision agriculture. It was well-structured and provided a valuable synthesis of a wide range of literature, particularly in detailing robotic components and available datasets for plant disease detection. The section on AI and thermal imaging was a notable strength, highlighting a promising research direction. However, the manuscript in its current form would benefit from significant revisions to enhance its analytical depth, clarity, and scholarly impact. The primary concerns are a need for deeper critical analysis, stronger integration between the three core technological themes(Robotics, AI and TI), and improved presentation of figures and tables. Detailed comments are as follows:

  • The current title accurately listed the three key technologies but presented them as parallel components. The manuscript's potential should be exploring their synergy. I suggest refining the title to better reflect this integrative focus. Correspondingly, the introduction and conclusion should more explicitly frame the review around how these technologies complement each other.

Response: Thank you for the review. Title, Section 1, Section 6.3 were updated.

  • There were 7 keywords in this manuscript, should be reduced.

Response: Thank you for your comment, keywords reduced.

  • Sections 3.1 and 3.2 provided a good descriptive overview of robotic end-effectors and platforms. However, a critical comparative analysis should be provided. A summary table comparing different mechanisms (e.g., success rates, damage rates, speed) and platforms (e.g., cost, terrain adaptability, payload) with both advantages and disadvantages should be listed.

Response: Thank you for the review. Section 3.1 and 3.3 were updated and tables were added.

  • While Section 6 effectively established the value of AI-powered thermal imaging, the rationale for this specific combination needed strengthening. The manuscript should more clearly articulate why thermal imaging is a uniquely powerful data source for AI models in agriculture compared to other sensing modalities, thereby justifying its focus.

Response: Thank you for the review . Section 5 was updated. Metrics section was removed so now section 6 is section 5

  • Figure 5, figure 10, figure 15 and figure 18 were not clear.

Response: Thank you for the comment. Figures replaced.

  • The order of Figures 13 and 14 should be reversed.

Response: Thank you for the comment. Figures where reversed

  • For Section 7, figure 16-18 were not cited in the main text, and the font size was wrong. The conclusion is too generic. It should synthesize the key findings for each technology area and, most importantly, provide a clear, forward-looking statement on the most promising pathways for integrating robotics, AI, and thermal imaging.

Response: Thank you for your comment. The figures were references and section was rewritten.

Reviewer 2 Report

Comments and Suggestions for Authors

We should thank the authors for a detailed and useful review! Comments on the article:
1. The overall structure of the review needs to be improved. The paper follows a logical sequence – from an introduction to a review of robotics, then AI, integration with thermal imaging, and discussion. However, some sections are loosely linked. For example, section 5 with definitions of metrics seems redundant. It simply lists the metrics Accuracy, Precision, Recall, and F1-score, although there are practically no new results (where these metrics are applied). This block does not carry a significant analytical load for the reader.
2. Insufficient completeness of the review of robotic solutions. The robotics sections focus primarily on harvesting robots and ground platforms. The text lists grippers and cutting mechanisms for tomatoes, strawberries, peppers, citrus fruits, etc.. However, other important areas of agricultural robotics have remained out of sight, such as specialized robotic seeders or sprayers. There is also no discussion of unmanned aerial vehicles (drones), which have already become a key tool for field monitoring and differentiated investment in precision agriculture (Guebsi, R.; Mami, S.; Chokmani, K. Drones in Precision Agriculture: A Comprehensive Review of Applications, Technologies, and Challenges. Drones 2024, 8, 686. https://doi.org/10.3390/drones8110686). For the declared "comprehensive" review, other types of robotic solutions should also be covered, including aerial robots and weed weeding/removal systems. Thus, the review needs to be expanded to cover tasks – not limited to harvesting, but to include all the basic operations of precision farming, where robotics is applicable.
3. The section devoted to AI includes detailed descriptions of neural network models, but does not sufficiently connect them with agro-practices. The authors examined in detail the well–known deep learning architectures – VGG16, InceptionV3, MobileNet - including their structure and development history. On the one hand, this demonstrates the depth of understanding of the authors, but on the other hand, it would be more important for an agronomist or engineer reader to find out how these models were applied in practice and how one is better than the other for specific tasks. In the review, the comparison of models is given only in the form of tables with percentages of classification accuracy on different datasets. The correctness and conclusions from the model comparison are not given enough. For example, the low accuracy (~12-16%) when transferring a model trained on PlantVillage to real PlantDoc photos highlights the problem of domain gap. The reader is forced to deduce it from the numbers himself. The review would benefit if the description of each model was followed by an analysis: what are its strengths and weaknesses specifically for agricultural tasks. For example, it should be noted that MobileNet, due to its small size, is suitable for implementation directly on the farm (in onboard devices) with minimal hardware, and VGG16 can be useful for additional training with a small amount of data, etc. 
4. The review of the AI application focuses only on the task of classifying diseases by images – other areas of agro-AI (weed detection, yield prediction, soil data analysis) are not mentioned. The AI review can be improved by providing a more applied analysis of models: not only their architecture, but also recommendations on which model is more appropriate for a particular agricultural scenario, based on accuracy, computational requirements, and data availability.
5. The analysis of sensor systems and mobile platforms is not critical enough. The review pays considerable attention to sensors and navigation: it lists a wide range of sensors (from power and ultrasonic to stereo cameras and LIDAR) and robot positioning methods. Tables with GNSS updates and reviews of sensor nodes for agriculture are presented, which confirms the authors' deep familiarity with the technical details. However, it would be useful to show how significant an improvement certain sensors provide for precision farming tasks – for example, to compare the effectiveness of stereo vision and LIDAR for orientation in rows of plants, or to discuss the limitations of inexpensive sensors (accuracy vs. accuracy). cost).
6. The Results and Discussion section unnecessarily repeats what was previously stated. In particular, sections 7.1–7.2 re-state that AI models successfully classify plant diseases, and thermal imagers detect pests – facts already mentioned in sections 4 and 6.

Author Response

Reviewer 2 Comments:

We should thank the authors for a detailed and useful review! Comments on the article:

2.1. The overall structure of the review needs to be improved. The paper follows a logical sequence – from an introduction to a review of robotics, then AI, integration with thermal imaging, and discussion. However, some sections are loosely linked. For example, section 5 with definitions of metrics seems redundant. It simply lists the metrics Accuracy, Precision, Recall, and F1-score, although there are practically no new results (where these metrics are applied). This block does not carry a significant analytical load for the reader.

Response: Thank you for your review. Section 5 was removed.

2.2. Insufficient completeness of the review of robotic solutions. The robotics sections focus primarily on harvesting robots and ground platforms. The text lists grippers and cutting mechanisms for tomatoes, strawberries, peppers, citrus fruits, etc.. However, other important areas of agricultural robotics have remained out of sight, such as specialized robotic seeders or sprayers. There is also no discussion of unmanned aerial vehicles (drones), which have already become a key tool for field monitoring and differentiated investment in precision agriculture (Guebsi, R.; Mami, S.; Chokmani, K. Drones in Precision Agriculture: A Comprehensive Review of Applications, Technologies, and Challenges. Drones 2024, 8, 686. https://doi.org/10.3390/drones8110686). For the declared "comprehensive" review, other types of robotic solutions should also be covered, including aerial robots and weed weeding/removal systems. Thus, the review needs to be expanded to cover tasks – not limited to harvesting, but to include all the basic operations of precision farming, where robotics is applicable.

Respones: Thank you for the review. The title of the paper was indeed misleading as I was more foucesd on how thesis technologies complement each in field of percison agriculture and thus I changed the title to “Fusion of Robotics, AI, and Thermal Imaging Technologies for Intelligent Precision Agriculture Systems” but I still discussed drones and what they offer to precision agriculture. I apologize for the overlook and thank you so much for the insight. UPDATES were made to section 2.2

2.3. The section devoted to AI includes detailed descriptions of neural network models, but does not sufficiently connect them with agro-practices. The authors examined in detail the well–known deep learning architectures – VGG16, InceptionV3, MobileNet - including their structure and development history. On the one hand, this demonstrates the depth of understanding of the authors, but on the other hand, it would be more important for an agronomist or engineer reader to find out how these models were applied in practice and how one is better than the other for specific tasks. In the review, the comparison of models is given only in the form of tables with percentages of classification accuracy on different datasets. The correctness and conclusions from the model comparison are not given enough. For example, the low accuracy (~12-16%) when transferring a model trained on PlantVillage to real PlantDoc photos highlights the problem of domain gap. The reader is forced to deduce it from the numbers himself. The review would benefit if the description of each model was followed by an analysis: what are its strengths and weaknesses specifically for agricultural tasks. For example, it should be noted that MobileNet, due to its small size, is suitable for implementation directly on the farm (in onboard devices) with minimal hardware, and VGG16 can be useful for additional training with a small amount of data, etc.

Response: Thank you for your review. Sections 4 and 6.3 were updated. Paragraphs were added that is more focused on how the models work in precision agriculture and their trade-offs. A comparison table was made.

2.4. The review of the AI application focuses only on the task of classifying diseases by images – other areas of agro-AI (weed detection, yield prediction, soil data analysis) are not mentioned. The AI review can be improved by providing a more applied analysis of models: not only their architecture, but also recommendations on which model is more appropriate for a particular agricultural scenario, based on accuracy, computational requirements, and data availability.

Response: Thank you for your review. Section 4 was updated, and a new table was added "Table 13".

2.5. The analysis of sensor systems and mobile platforms is not critical enough. The review pays considerable attention to sensors and navigation: it lists a wide range of sensors (from power and ultrasonic to stereo cameras and LIDAR) and robot positioning methods. Tables with GNSS updates and reviews of sensor nodes for agriculture are presented, which confirms the authors' deep familiarity with the technical details. However, it would be useful to show how significant an improvement certain sensors provide for precision farming tasks – for example, to compare the effectiveness of stereo vision and LIDAR for orientation in rows of plants, or to discuss the limitations of inexpensive sensors (accuracy vs. accuracy). cost).

Response: Thank you for the review. Updates were made to section 2.5. I discussed the limitations of the sensors and the trade-off between accuracy and cost.

2.6. The Results and Discussion section unnecessarily repeats what was previously stated. In particular, sections 7.1–7.2 restate that AI models successfully classify plant diseases, and thermal imagers detect pests – facts already mentioned in sections 4 and 6.

Response: Subsections 7.1 and 7.2 were removed as they were redundant.

Reviewer 3 Report

Comments and Suggestions for Authors

Line 4: “This paper focuses on analyzing the changes brought to the contemporary agricultural value chain…”
→ Do you mean a systematic review of value-chain impacts, or mainly technical performance of robots/AI/TI? The rest of the abstract reads like a tech review, not a value-chain analysis (inputs, logistics, markets).

Line 12: “Among the losses, AI-TI integration recognizes early temperature fluctuations…”
→ The phrase “Among the losses” doesn’t parse. Do you mean to reduce losses?

Line 14: “inconsistency of the environment” → Do you mean environmental variability (illumination, wind, canopy water status) that breaks models?

Line 28: “This is further demonstrated by the United Nations with SDG2… 50×2030 Initiative [3].”
→ “Demonstrated” feels off. SDG2 is a goal, not evidence. Also, how does 50×2030 tie to precision agriculture data specifically? One bridging sentence is missing.

Line 34:  “For more than three decades [6], agricultural robotics has been studied…”
→ OK, but are you reviewing field-validated robots or also lab prototypes? The scope isn’t explicit.

Line 35:  “These Robots substitute for the labor shortage…”
→ Capitalization of Robots

Line 42:  TI section: “TI can show subtle temperature variations linked to early-stage infestations and infections [9].”
→ I need the assumptions: emissivity setting, time-of-day standardization, wind and VPD control. Without that, canopy temperature can reflect water status more than disease.

Line 59:  Metrics paragraph: “accuracy, precision, recall, F1” → Good, but field use also needs latency, FPS, model size, edge inference feasibility. Are you going to cover those or not?

Line 72:  “The objective is to demonstrate how smart systems can enhance… and optimize resource utilization…”
→ Are you demonstrating with new experiments, or synthesizing literature? The wording blurs review vs original research.

Line 77:  “It will also discuss future work…”
→ What gaps are you committing to analyze (label scarcity, domain adaptation, multimodal fusion, economic ROI)?

Line 142:  You classify platforms (railed, tracked, four-wheel, independent steering). I’m missing a task–terrain matrix: which platform for mud, slopes, row spacing <70 cm, canopy height limits?

Line 161: Power: you note “24 V DC battery for 2–3 h”. At what duty cycle and payload? Field coverage (ha h⁻¹) would help.

Figure 2(e) “Husky A200” says it fits cotton row spacing (68 cm width). How is row-centering achieved (vision vs GNSS vs LiDAR)?

Line 181: You say reliable, accurate, low-cost sensors are a challenge—agree—but then jump to GNSS upgrades (Table 2). For row-crop under canopy, GNSS often degrades. How do systems fuse GNSS with vision/IMU?

Line 193: YOLOv3 mentioned for object detection. Why v3 and not more recent YOLOv5/8 or transformers? Is this deliberate (edge compute constraints)?

Line 220:  ROS: you list “five main modules.” ROS doesn’t ship modules per se; do you mean stack components (perception, localization, planning, control, navigation)? Also, are you using ROS 1 or ROS 2 (DDS, real-time)?

Line 271:  mAP = 20.2 claim: give the exact baseline (model, backbone, anchors, input size), train/val/test split, evaluation protocol (IoU threshold, per-class AP), and confidence intervals. If this is from a cited paper, say so explicitly.

Line 309:  Inconsistency: you first say PlantDoc “same goes … gathered from the internet,” then L305–321 correctly state FieldPlant is farm-captured. Reword to avoid implying PlantDoc is internet-only; cite the original PlantDoc paper’s data sources.

Line 440:  Define TP/TN/FP/FN right above the equations (you reference them but the symbols first appear in the formulas). State whether metrics are macro- or micro-averaged for multi-class datasets.

Line 462:  Balance the advantages with limitations: emissivity differences, canopy angle, sun/sky reflections, wind, sensor drift, and need for field calibration/ground truth. Add 1–2 citations.

Line 511: Post-harvest TI: you report error margins 0.410 and 0.086 without units—are these RMSE in °C? Please specify metric and dataset; briefly describe the setup (materials, container types, ambient control). The “200 °C cauterization” example needs safety/context (species, exposure time, why applicable to agriculture).

Line 519: “Fusion of thermal and visible… outperformed traditional methods like LPT.”

→ Define LPT at first use; add quantitative benchmark (accuracy/F1/ΔRMSE) and dataset/task so “outperformed” is verifiable.

Line 541: Section title “Results and Discussion” but content is a narrative summary.
→ Either present new results (tables/figures) or rename to “Synthesis and Implications.

Author Response

Reviewer 3 Comments:

Line 4: “This paper focuses on analyzing the changes brought to the contemporary agricultural value chain…”
→ Do you mean a systematic review of value-chain impacts, or mainly technical performance of robots/AI/TI? The rest of the abstract reads like a tech review, not a value-chain analysis (inputs, logistics, markets).

Response: Thank you for the review. The main focus is indeed the technical performance and how they impact precision agriculture. Abstract was updated to remove any ambiguity.

Line 12: “Among the losses, AI-TI integration recognizes early temperature fluctuations…”
→ The phrase “Among the losses” doesn’t parse. Do you mean to reduce losses?

Response: Thank you for the review. Yes. Abstract was updated.

Line 14: “inconsistency of the environment” → Do you mean environmental variability (illumination, wind, canopy water status) that breaks models?

Response: Thank you for the review. Yes. Abstract was updated.

Line 28: “This is further demonstrated by the United Nations with SDG2… 50×2030 Initiative [3].”
→ “Demonstrated” feels off. SDG2 is a goal, not evidence. Also, how does 50×2030 tie to precision agriculture data specifically? One bridging sentence is missing.

Response: Thank you for the review. Section 1 was updated to fix this problem.

Line 34:  “For more than three decades [6], agricultural robotics has been studied…”
→ OK, but are you reviewing field-validated robots or also lab prototypes? The scope isn’t explicit.

Response: Thank you for the review. Line 34-36 were updated. We reiewing all advances in the robotics field

Line 35:  “These Robots substitute for the labor shortage…”
→ Capitalization of Robots

Response: Thank you for the review. Line 36 ”Robots” was capitalized

Line 42:  TI section: “TI can show subtle temperature variations linked to early-stage infestations and infections [9].”
→ I need the assumptions: emissivity setting, time-of-day standardization, wind and VPD control. Without that, canopy temperature can reflect water status more than disease.

Response: Thank you for the review. Introduction was updated to remove ambiguity.

Line 59:  Metrics paragraph: “accuracy, precision, recall, F1” → Good, but field use also needs latency, FPS, model size, edge inference feasibility. Are you going to cover those or not?

Response: Thank you for the review. This entire section was removed as it doesn’t significantly contribute to the scope of the paper or the journal.

Line 72:  “The objective is to demonstrate how smart systems can enhance… and optimize resource utilization…”
→ Are you demonstrating with new experiments, or synthesizing literature? The wording blurs review vs original research.

Response: Thank you for the review. Introductioin was updated to remove ambiguity. This is a literature review indeed.

Line 77:  “It will also discuss future work…”
→ What gaps are you committing to analyze (label scarcity, domain adaptation, multimodal fusion, economic ROI)?

Response: Thank you for the review. Introduction was updated. Domain adaptation should be in future work to fix domain gap.

Line 142:  You classify platforms (railed, tracked, four-wheel, independent steering). I’m missing a task–terrain matrix: which platform for mud, slopes, row spacing <70 cm, canopy height limits?

Response: Thank you for the review. Section 3.2 was updated and a new table was created as a task terrain matrix.

Line 161: Power: you note “24 V DC battery for 2–3 h”. At what duty cycle and payload? Field coverage (ha h⁻¹) would help.

Response: Thank you for the review. Section 3.2 was updated an approximation was made as it was not explicitly made clear what was the duty cycle and field coverage.

Figure 2(e) “Husky A200” says it fits cotton row spacing (68 cm width). How is row-centering achieved (vision vs GNSS vs LiDAR)?

Response: Thank you for the review. Section 3.2 was updated to fix ambiguity.

Line 181: You say reliable, accurate, low-cost sensors are a challenge—agree—but then jump to GNSS upgrades (Table 2). For row-crop under canopy, GNSS often degrades. How do systems fuse GNSS with vision/IMU?

Response: Thank you for the review. Section 3.3 was updated. To show how to mitigate this problem which is sensor fusion

Line 193: YOLOv3 mentioned for object detection. Why v3 and not more recent YOLOv5/8 or transformers? Is this deliberate (edge compute constraints)?

Response: Thank you for the review. Yes, this was deliberate, as it aligns with the same cited work from which the entire section was reviewed to ensure consistency and reliability as the author used and presented results for YOLO3

Line 220:  ROS: you list “five main modules.” ROS doesn’t ship modules per se; do you mean stack components (perception, localization, planning, control, navigation)? Also, are you using ROS 1 or ROS 2 (DDS, real-time)?

Response: Thank you for the review. Section 3.3 was updated. In the original cited work ROS 1 was being used.

Line 271:  mAP = 20.2 claim: give the exact baseline (model, backbone, anchors, input size), train/val/test split, evaluation protocol (IoU threshold, per-class AP), and confidence intervals. If this is from a cited paper, say so explicitly.

Response: Thank you for the review. Section 4.1.1 was updated.

Line 309:  Inconsistency: you first say PlantDoc “same goes … gathered from the internet,” then L305–321 correctly state FieldPlant is farm-captured. Reword to avoid implying PlantDoc is internet-only; cite the original PlantDoc paper’s data sources.

Response: Thank you for the review. Section 4.1.4 was updated.

Line 440:  Define TP/TN/FP/FN right above the equations (you reference them but the symbols first appear in the formulas). State whether metrics are macro- or micro-averaged for multi-class datasets.

Response: Thank you for the review. Whole section was removed as it does not contribute much to the scope of the paper or journal.

Line 462:  Balance the advantages with limitations: emissivity differences, canopy angle, sun/sky reflections, wind, sensor drift, and need for field calibration/ground truth. Add 1–2 citations.

Response: there is no comparison in this section

Line 511: Post-harvest TI: you report error margins 0.410 and 0.086 without units—are these RMSE in °C? Please specify metric and dataset; briefly describe the setup (materials, container types, ambient control). The “200 °C cauterization” example needs safety/context (species, exposure time, why applicable to agriculture).

Response: Thank you for the review. Section 5.2 was updated.

Line 519: “Fusion of thermal and visible… outperformed traditional methods like LPT.”

→ Define LPT at first use; add quantitative benchmark (accuracy/F1/ΔRMSE) and dataset/task so “outperformed” is verifiable.

Response: Thank you for the review. Section 5.2 was updated.

Line 541: Section title “Results and Discussion” but content is a narrative summary.
→ Either present new results (tables/figures) or rename to “Synthesis and Implications.

Response: Thank you for the review. Section title was updated

Round 2

Reviewer 1 Report

Comments and Suggestions for Authors

Some of the pictures were still not clear, such as figure 5 and figure 10. The text on the pictures can't be seen clearly even if you enlarge it.

Author Response

Thank you so much for your comments

Both figures have been recreated

Reviewer 2 Report

Comments and Suggestions for Authors

The adjustments made have significantly improved the article.

Author Response

Thank you so much